# Vitamin K and the Visual System—A Narrative Review

**DOI:** 10.3390/nu15081948

**Published:** 2023-04-18

**Authors:** Michael A. Mong

**Affiliations:** Department of Ophthalmology, Veteran Affairs North Texas Health Care Medical Center, Dallas, TX 75216, USA; michael.mong@va.gov

**Keywords:** vitamin K, vitamin K1, vitamin K2, eye, visual function, cognitive function, vitamin K-dependent protein, matrix Gla protein, ferroptosis

## Abstract

Vitamin K occupies a unique and often obscured place among its fellow fat-soluble vitamins. Evidence is mounting, however, that vitamin K (VK) may play an important role in the visual system apart from the hepatic carboxylation of hemostatic-related proteins. However, to our knowledge, no review covering the topic has appeared in the medical literature. Recent studies have confirmed that matrix Gla protein (MGP), a vitamin K-dependent protein (VKDP), is essential for the regulation of intraocular pressure in mice. The PREDIMED (Prevención con Dieta Mediterránea) study, a randomized trial involving 5860 adults at risk for cardiovascular disease, demonstrated a 29% reduction in the risk of cataract surgery in participants with the highest tertile of dietary vitamin K1 (PK) intake compared with those with the lowest tertile. However, the specific requirements of the eye and visual system (EVS) for VK, and what might constitute an optimized VK status, is currently unknown and largely unexplored. It is, therefore, the intention of this narrative review to provide an introduction concerning VK and the visual system, review ocular VK biology, and provide some historical context for recent discoveries. Potential opportunities and gaps in current research efforts will be touched upon in the hope of raising awareness and encouraging continued VK-related investigations in this important and highly specialized sensory system.

## 1. Introduction

Losing one’s sight consistently ranks first among the most feared conditions, with blindness seen as the worst possible affliction among a list that includes Alzheimer’s disease (AD), AIDS/HIV, cancer, loss of limb, heart disease, and deafness [1]. As early as 1525 B.C., the Ebers Papyrus highlights this long-standing human concern regarding vision loss by chronicling numerous eye diseases and treatments, many of which were diet-based [2], and Hippocrates appreciated the importance of the eye as a biomarker and prognostic indicator circa 500 B.C. [3,4].

Blindness in the 21st century remains an enormous challenge as the global population continues to grow and age. Age-related, as well as behavioral and lifestyle-associated conditions, such as glaucoma, cataracts, age-related macular degeneration, and diabetic retinopathy, are assuming greater importance, as eye care services and treatments cannot keep up with demand despite numerous advances [5,6]. The best estimates of global blindness in the year 2020 suggest that 43.3 million people are blind and 295 million have moderate to severe vision loss, and by 2050, the numbers are predicted to increase to 61 million and 474 million, respectively [5]. The most recent analysis of the burden of significant visual acuity loss and blindness (VL) in the US population indicates the problem may be much more significant than previously appreciated [7]. The mean prevalence of VL was 2.17% for the year 2017, which increased to 20.37% in persons aged 85 years and older, representing nearly 7.08 million people, 1.08 million of whom have blindness, a 68.7% increase over previous estimates [7]. The cost of VL is significant, with estimates of the yearly US total economic burden of VL in 2017 being USD 134.2 billion—USD 98.7 billion direct, and USD 35.5 billion in indirect costs [8]—and the intangible price that blindness exacts from those afflicted, and their families, is also substantial [9].

Experts acknowledge that nutrition and dietary interventions are the most powerful modifiable risk factors for disease across the life span, and correction of micronutrient deficiencies can represent a substantial return on investment for individuals and society [10]. It is also recognized that it is imperative to understand the nutritional factors that can drive the prevention and treatment of age-related conditions [11]. Though evidence exists that nutritional inputs can delay diseases such as macular degeneration and cataracts, public health measures related to functional losses are lacking, and fresh approaches to conduct both rapid and rigorous dietary interventions in diverse populations are needed [12]. The relative lack of research with respect to vitamin K in general [13], and with respect to the visual systems in particular, is striking and deserving of particular attention.

Not a single trial in the US National Institutes of Health ClinicTrial.gov database was found related to VK and the eye or visual systems [14], and there is only one trial in the National Library of Medicine PubMed.gov database related to vitamin K1 and the reduction in cataract surgery risk [15]. Thus, the opportunity to further our understanding of the VK biology and specific VK requirements of the visual system is evident, with reports that for each dollar spent on vision loss prevention and treatment, there is at least a USD 5 return on investment [16].

Vitamin K is a fat-soluble vitamin whose severe deficiency can lead to potentially fatal bleeding and blindness [17], a fact which has dominated a great deal of its history, but whose long-term deficiency has recently been linked with a number of clinical conditions associated with aging [18]. VK is one of four known fat-soluble vitamins, which also includes vitamins A, D, and E. In sharp contrast to its fellow vitamins, VK is neither stored in the body to any great degree, nor is it produced de novo, with plasma concentrations of phylloquinone (PK or vitamin K1) around 0.5 nM, which is 1–4 orders of magnitude less than 25-hydroxyvitamin D, retinal (a form of vitamin A), and α-tocopherol (a form of vitamin E) [19]. Consequently, its physiologic function is predicated upon the ongoing and critical intracellular regeneration of its active form, vitamin K hydroquinone (VKH2), dietary intake, gut microbiome synthesis, or therapeutic supplementation [19]. The pool size of PK in healthy volunteers may be as low as 17–194 μg (0.28–2.17 μg/kg), with a mean and (SD) of 87.6 μg (60), and a body pool turnover of approximately 1.5 days [20], highlighting the potentially precarious nature of VK metabolism within the human body.

VK is a general term for 2-methyl-1,4-naphthoquinone or menadione (MD, K3) and any of its derivatives with antihemorrhagic effects [21]. MD is a synthetic form of VK used in early work, and its water-soluble derivatives remain an important addition to animal feed, particularly in chickens (and thus to the human food chain), who are acutely vulnerable to VK-related deficiency and bleeding. MD is now known, however, to be an important endogenously produced intermediary in human VK metabolism as a source of menaquinone-4 (MK4) [21]. As isoprenoid quinones, these compounds are among the most important in living organisms as a result of their ability to share electrons and participate in redox cycles [22], with menaquinones (MK, MK-n) functioning as part of the respiratory electron transport chain in bacteria, and phylloquinone (PK) participating in electron transport for photosynthesis in plants [23].

The VK isoprenoid quinones are highly conserved evolutionally and their metabolism is complex compared to its fellow fat-soluble vitamins in important ways which remain under intense investigation. The reader is referred to the excellent reviews by Shearer et al., 2014, 2018 [19,24]. It has long been believed that the only firmly established function of VK is its action as a cofactor for a single-membrane-bound endoplasmic reticular enzyme, γ-carboxyglutamyl carboxylase (GGCX), and its central role in maintaining hemostasis via the liver is its only health role supported by incontrovertible evidence [19]. However, in addition to supporting hemostasis, extrahepatic vitamin K-dependent proteins (VKDP) are now known to impact a broad range of processes of potential importance to the visual system, including the regulation of tissue calcification, apoptosis, growth control, signal transduction, and angiogenesis [25]. In addition, non-canonical VK pathways, unrelated to VKDP, such as the nonenzymatic, antioxidant-mediated inhibition of ferroptotic programmed cell death [26], and the activation of the steroid and xenobiotic (SXR) nuclear hormone receptor [27], are being shown to have potentially significant roles in ocular health [28].

The synthesis of VKDPs by GGCX appears to be a unique biochemical process that generates a previously unknown amino acid, γ-carboxyglutamic acid (Gla), with two negative charges, from VKDP glutamic acid (Glu) residues. It was first reported by Stenflo and Nelsestuen in 1974 [29], as part of their effort to discover the structure and function of prothrombin [30]. This conversion adds an additional negative charge to the Glu residues, which allows for a unique calcium ion (Ca^++^)–protein interaction that had not been previously described [30].

Calcium is the third most abundant metal in nature and Ca^++^ ions are the most tightly regulated in membrane-bound organisms and organelles, with a 20,000-fold gradient between the intracellular (~100 nM) and extracellular (mM) concentrations [31]. Cells, and the organelles within them, such as the mitochondria and endoplasmic reticulum (ER), must chelate, compartmentalize, or extrude these ions by binding them to proteins to influence function, localization, and association [31]. Seventeen currently known VKDPs represent such highly specialized proteins in humans [32,33]. Protein function is determined by shape and charge, and Ca^++^ binding, as well as phosphorylation, triggers these changes in proteins and thus regulates their function [31]. In nature, six to seven oxygen atoms of carboxyl groups surround and coordinate a single Ca^++^ ion, forming a pentagonal bipyramid [31,34], changing the VKDP conformation, and thus affecting its activation.

A detailed examination of VK metabolism, VK-related molecular pathways, VKDP activation, and the VK content of common foods is beyond the scope of this work, and the reader is referred again to the reviews by Shear et al. (2014, 2018) [19,24], and a comprehensive review on VK metabolism and VK metabolic pathways by Beato et al. (2020) [35], along with the recent detailed coverage of VKDP activation via GGCX by Berkner and Runge (2022) [25]. Additionally, Schurgers and Vermeer in 2000 [36] and Elder et al. in 2006 [37] provide detailed analyses of the PK and Mk-n content of common foods in the Netherlands and the United States, respectively.

The question of what constitutes, and how best to obtain, an adequate intake of VK for the EVS is anything but settled [38,39,40]. It is known that many people suffer from the lack of minute amounts of various trace nutrients, with adverse consequences that can include poor health, mental impairment, and even blindness—the prevention of which, in theory, can be cheap and simple, but in practice has been found to be a complex endeavor, and often met with failure [41]. Increasingly, it is being recognized that traditional nutritional epidemiological approaches have significant limitations [42], with a call to integrate recent developments in genomic, epigenomic, transcriptomic, proteomic, microbiome, and metabolomic information into a “systems epidemiologic” approach to support human nutritional needs over the entire life cycle from gestation through adult life [43]. The goal of such an approach is to move beyond dietary recommendations based on population averages which may not be optimal towards precision nutrition and personalized nutritional recommendations and interventions [44]. These approaches are likely to be of particular importance with respect to VK, as it has been shown to have markedly greater intra-individual and interindividual variation in circulating plasma levels compared to any of the other FSVs [45] and large, genetically associated interindividual responses to dietary intake and supplementation [46].

Current recommendations with respect to VK intake are framed as Adequate Intake (AI) values for PK and have been set at 120 μg per day for men and 90 μg for women in the USA [47] and at 70 μg/day for both men and women over the age 19 years in the EU [48]. The importance of menaquinones from meat, eggs, and dairy products is being increasingly recognized [36,37], as are fermented foods such as natto and other similar products in Asia [49].

Table 1 provides an abbreviated list of common foods with higher relative concentrations of PK and MK4 obtained from the USDA FoodData Central database [50,51].

Increasingly, as nutrient databases expand with respect to the MK-n content of foods, the relative proportions and sources of PK and MK-n in contemporary diets are becoming better understood. For example, in 2022, reporting on a total of 1985 Polish adults aged 35–70 years, Regulska-Ilow et al. [52] found that females had a mean and SD total VK intake of 358.6 ± 181.0 (μg/Day) composed of 205.2 ± 138.2 PK and 153.4 ± 87.7 MK-n, while males had mean and SD intakes of 331.1 ± 151.5, 164.5 ± 94.9, and 166.6 ± 92.0, respectively. This represents a PK:MK-n ratio of 1.34 for females and 0.99 for males. Vegetables, processed meats, high-fat cheeses, soups, and low-fat dairy represented the top five food groups and comprised 77.5% of the total VK in females. In males, the top five food groups were vegetables, processed meats, high-fat cheeses, soups, and red meats, which was 76.5% of the total VK in the diet. Figure 1 provides a visual summary of these data.

Statistically significant differences were found between females and males in terms of relative PK, MK-n, and total VK in their diets and among urban and rural residents. Mean ± SD PK and total VK was higher in women, while men had higher MK-n intake. Urban women had higher PK and total VK intake than rural women, while rural women were higher in MK-n. Urban males were higher in PK, while rural males were higher in both MK-n and total VK. Figure 2 provides a summary of these findings.

## 2. Vitamin K Biology and the Visual System

Limited data exist from which we can begin to understand VK metabolism, transport, and pathways within the EVS. As visual processing is estimated to occupy 55% of the primate neocortex [53] and is one of the higher cortical functions of the human brain [54], brain-related VK data are included as important relevant information within this review.

Some of the fundamental questions to be answered are: (1) What is the nature of VK action in the eye, and by extension, how might this compare with the brain and central nervous system in general? (2) How is VK trafficked in the EVS? (3) What constitutes optimized VK levels in the EVS, and how is this best achieved and maintained over the life span? and (4) What new biomarkers related to VK status might be present and readily accessible in the EVS? None of the answers to these questions are definitively known.

### 2.1. Vitamin K Content and Delivery to Ocular and Related Tissue

Only one study reports the tissue concentration of PK and MK4 in the eyes of female and male mice [55]. The remainder of the data must be extrapolated from reports of PK and MK content and their distributions in human brains, as well as animal brain studies, or gleaned from analyses of associated factors in human and bovine vitreous and aqueous humor. Studies related to the delivery of other nutrients to the eye and brain via LDL and HDL also exist.

In their 2008 foundational work, Okano and co-workers [55] used deuterium labeling to provide unequivocal evidence that MK4-d7 in the mouse cerebrum originates from dietary PK-d_7_ and/or MD-d_8_ by one of two routes, either by the conversion of PK to MD in the intestines, followed by MD uptake and conversion to MK4 in the cerebrum, or uniquely, tissue-specific PK > MD > MK4 conversion within the brain; evidence of the latter route was presented for the first time in this study. They concluded that their data suggest that the cerebrum is particularly unique as it contains relatively large amounts of MK4 but little to no PK, likely the result of the blood–brain barrier (BBB) or other unknown mechanisms [55]. They also provided valuable oral PK > MD > MK4 kinetic data showing that, in female mice, PK-d_7_ and MK-d_8_ were converted into cerebral MK4-d_7_ over 24 h in a dose-dependent fashion from a physiological dose of 0.1 μmol/kg body weight to what was considered a pharmacological dose of 10 μmol/kg body weight. Additionally, they showed that a single oral administration of PK-d_7_ at 10 μmol/kg body weight resulted in the accumulation of MK-4-d_7_ in the cerebra at a concentration of 82.9 ± 6.1 pmol/g tissue, nearly 30% of the concentration (252.5 ± 10.3) of MK-4 in the cerebra of age-matched female mice within 24 h [55].This conversion of PK to MD had long been understood to occur but was not previously proven. Many aspects of this pathway remain important open questions.

Table 2 below appears to be the only source of detailed information documenting and contrasting VK levels in the eyes and 27 other tissues of mice. There is variability in the PK and MK4 concentrations among tissue types, and even among different regions of the brain, which is in concert with MK/PK concentration data reported in nine separate areas of the rat brain, but without eye-associated information, reported previously by Ferland in 2004 [56].

Importantly, the eye was found to contain both PK and MK4 with an MK4/PK ratio of 6.75 and 5.63 in females and males, respectively, and appears to be enriched with respect to PK compared to the cerebrum and to be reduced in MK4 with ratios of 0.41 and 0.55 in females and males, respectively. Ocular PK and MK4 concentrations are higher in females with a ratio of 1.5 for PK and 1.8 for MK4. Though the *p* values comparing female and male tissue concentrations were not presented in the original publication, the raw eye data were graciously provided by Professor K. Nakagawa (Personal communication 6 March 2023), who indicated that intracellular expression of UBIAD1, the enzyme responsible for the biosynthesis of intracellular MK4, is largely the cause of differences in MK4 levels among the sexes; however, the significance of higher MK4 concentrations in females is not clear. Interestingly, utilizing a Mann–Whitney U test, female sex was found to be associated statistically with MK4 concentration in the eye with a value *p* = 0.01, while PK was not, with a value of *p* = 0.30. PK values appear less certain given smaller concentrations and larger SE values. Caution should be taken in drawing any conclusions from these preliminary ocular data, however, as the primary purpose of the study was to understand the generation of MK4 within the mouse cerebrum. The eye, in part an embryologic extension of the brain, is, however, less homogeneous than the brain, containing many different tissue types, including a large volume of vitreous and aqueous humor which is 99% water, providing a rationale for undertaking an aggressive program to map tissue-specific MK4 (and possible other MKn) and PK concentrations of the human eye.

Figure 3 below is a graphical presentation of the tissue VK concentrations (pmol/g tissue or pmol/mL plasma) in mice fed a normal diet sorted with respect to female MK4 tissue levels adapted from the data in Table 2, along with plots of the MK4 and PK eye concentrations compared according to sex.

Building upon their previous work, in 2013, Nakagawa, Hirota, and colleagues [57] unequivocally demonstrated using deuterium-labeled PK (PK-d_7_) that tissue MK4 in rats is primarily derived from dietary PK that is converted to MD in the intestines and released into general circulation by way of the lymphatic and vascular circulatory systems. They further demonstrated a similar time course change in PK, MK4, and MD concentrations in healthy adult volunteers, which they believed provided evidence of similar action in humans. Their kinetic data suggested that the bulk of the PK was absorbed in the intestine and passed unaltered via the mesenteric circulation to the liver, but that a smaller, yet critical, amount of PK was converted to both MD and MK4 in the intestines and entered the mesenteric lymphatics. The lymph concentrations of PK-d_7_ were approximately 50-fold higher than those of MD-d_7_ and 1000-fold higher than those of MK4-d_7._ They maintain that their tissue concentration data, along with their previous work [55], indicate that PK is mainly stored in the liver, and that the cerebrum actively accumulates MK4 in rats and mice, with the accumulations in mice being higher than those in rats. Unfortunately, no ocular data were provided in this work.

The authors comment that their results are in concert with Thijssen et al. (2006) [58], who found that approximately 5–20% of an oral dose of PK in healthy male volunteers was catabolized to MD and that unsupplemented subjects had a daily excretion of MD in the urine of 1.6–9.1 μg. The implication for the eye and brain are clear, as both Nakagawa and Thijssen note, that the smaller lipophilic MD can more easily pass the blood–brain barrier (BBB), and by extension, the blood–retinal barrier (BRB), to supply intracellular MK4 upon conversion. Partridge (1998) [59] and Vita et al. (2011) [60] detail that lipid-soluble molecules with a mass under 400–600 Da are transported readily by lipid-mediated free diffusion through the vascular endothelial plasma membranes and are desirable for drug delivery in the central nervous system, which may indicate that MK is able to enter the circulation rapidly and passively after conversion in the enterocytes. The molecular weight of MD is 172.2 Da, and by contrast, MK4 is 444.6 Da, PK is 450.07 Da, MK7 is 649.0 Da, MK9 is 785.23 Da, and MK13 1057.7 Da. It is important to note that this property of MD may be a double-edged sword in that the use of MD as a pharmacologic agent to treat VK deficiency was stopped long ago, as it caused hemolytic anemia and liver damage [21], but it is now being used in small, titrated doses along with high-dose vitamin C to treat glioblastoma [60]. Thus, as recently demonstrated by Ellis et al. (2021) [61], there appears to be a very elegant sorting mechanism by which abundant PK, and possibly all dietary MKn forms, are converted to smaller, more lipid-soluble MD in a titrated manner to facilitate delivery to the brain and eye, while larger forms are packaged into chylomicrons, LDL, and HDL for export via vascular circulation to the liver, pancreas, kidney, bones, eye, brain, and other sites.

With respect to human studies, Thijssen et al. (1996) [62] reported on six autopsy cases, three of which contributed brain tissue. PK was found in all tissue types, with relatively small amounts in the brains with a median and (SD) of 1.5 (0.7) pmol/g. MK4 levels were relatively high in the brain, with a median of 6.3 (2.8) pmol/g. MK4 exceeded PK levels only in the brain and kidney. The MK4/PK ratios in the brains were 2.4, 3.4, and 13.3, with a mean of 6.4 (6.0) (not unlike the MK4/PK ratios of 5.63–6.75 reported by Okano in 2008 in mouse eyes). In contrast, the livers were found to have total VK stores ranging between 40 and 125 pmol/g wet weight, with PK ranging from 4 to 45% of the total. Thus, both PK and MK were consistently found in the human brain.

In 2021, Tanprasertsu et al. [63] reported on the MK4 and PK levels in the frontal and temporal cortex of 45 decedent centenarians in the Georgia Centenarian Study with and without dementia. The overall mean (SD) concentrations for MK4 were 4.96 (2.32) and 0.40 (0.39) pmol/g for PK, which were not significantly different between those with and without dementia. Of interest and potential significance, all brains were found to contain MK4, but in 38%, no PK was detected. The MK4/PK ratio was 12.4 among all subjects, which is within the range reported by Thijssen. Interestingly, in a companion paper reporting on the same individuals [64], the mean pmol/g and (SD) brain retinol levels were 691.97 (305.56) and the mean pmol/g and (SD) α-tocopherol levels were 66,917 (13,676). The ratios of retinol to MK4 and PK were thus 140 and 1730, respectively, and those to α-tocopherol were 13,491 and 167,293, both highlighting and reaffirming the low relative tissue levels of VK compared to other FSVs [19].

Further brain VK data were found in a retrospective analysis of frozen brain tissue. Fu et al. (2021) [65] report on 499 participants in the Rush Memory and Aging Project (mean age 92 y, 72% female) and found the MK4 level in brain tissue stored from 0.4 to 13.0 years was 1.2 pmol/g, which they noted was much lower than what others have found. PK levels were reported to be non-detectable, though small amounts were found below levels able to be accurately reported.

In a recent major advancement, in 2022, Booth et al. [66], reporting on a subset of the Rush Memory and Aging Project, presented data linking VK content in 325 human brains, for the first time, to post-mortem neuropathologic histology and cognitive function prior to death. Brain MK4 values, presented as mean pmol/g (SD), were found to be 1.51 (5.5), 0.73 (4.6), and 1.61 (7.2) in the mid-frontal/mid-temporal cortex, anterior watershed area, and cerebellum, respectively. Brain MK4 concentrations were inversely correlated with global Alzheimer’s disease (AD) pathology, neurofibrillary tangle density, Braak stage, and Lewy body presence (all well-established markers of neurodegeneration), as well as positively correlated with higher cognitive function and a slower rate in cognitive decline prior to death. Both the neurodegenerative findings and cognitive decline have clear connections to visual function and clinically relevant anatomic correlations to ocular findings, which will be addressed subsequently. Notably, the average brain MK4 concentrations, which were very carefully assayed and validated, were found to be three to five times lower than those reported by Tanprasertsu et al. [63] and Thijssen et al. [62], which may reflect differences in the way the brains were handled and stored, as well as in assay protocols, or, possibly, actual variations among the populations.

Thus, the available data suggest that MK and PK are consistently found in human, mouse, and rat brain tissues with MK4/PK ratios that vary but consistently favor MK4. Mouse evidence documents MK4 and PK in the eye with MK4/PK ratios of 6.75-5.63 in females and males, respectively. MK4 and PK levels in human eyes are currently unreported. Furthermore, a portion (possibly 5–20%) of dietary PK, MK4, and likely MK-n forms as well are converted to MD in the intestines and passively absorbed into the circulation because of their small M.W (172.2 Da) and lipophilicity, which are key physicochemical properties determining the absorption, distribution, metabolism, excretion, and toxicity (ADMET) of a compound [67]. MD may then rapidly cross the BBB and BRB to be taken up by the cells in the brain and eye and quickly converted to MK4. In addition to providing an elegant mechanism to overcome the important BBB and BRB, and likely optimize brain and ocular VK levels, this PK > MD > MK4 conversion has long suggested to experts [19,21,57,61,62] that MK4 likely has important intracellular functions in addition to the carboxylation of VKDP’s.

Corollaries to the above discussion are the facts that 75% or more of dietary PK and MK is not converted to MD and that the conversion of PK > MD > MK4 does not explain the mechanism for the accumulation of PK in the eye and brain. PK and MK, as lipophilic vitamins, along with cholesterol and carotenoids such as lutein and zeaxanthin, are insoluble, non-swelling, amphiphilic polar lipids absorbed along with other dietary fats, as has been expertly reviewed by Ji et al. in 2009 [68], again by Shearer et al. in 2012 [69], and more recently by Lai et al. in 2022 [70]. Important with respect to the ocular system is the fact LDL and HDL each carry approximately 7% of serum-associated PK [71], with LDL containing both MK4 and MK9 forms as well, while HDL was found to carry only MK4 [72], which may be particularly important with respect to preferential MK4 concentration in the eye and brain.

In the early 1980s, it was pointed out that lipids were present in human aqueous humor (AH) at a mean concentration of 16.4 mg/dL, with HDL likely to be the sole lipid-transporting lipoprotein, at concentrations of 4 micrograms/mL. This suggested to the authors that HDL was transported into the anterior chamber via AH and that HDL may be an important source of lipids to the lens and other anterior chamber tissues [73,74]. The near exclusive and relatively abundant presence of HDL-related apoproteins in the AH, and that of the only rare or absent LDL-related ApoB lipoprotein, has been confirmed using state-of-the-art liquid-chromatography mass spectrometry (LC-MS/MS) by two groups in 2009 and 2020 [75,76]. Furthermore, Conner et al. (2007) [77] and Li et al. [78] have demonstrated the unique role HDL plays in the delivery of carotenoids to the retinal pigment epithelium and retina, and likely by extension MK4, as demonstrated by Schurgers and Vermeer 2002 [72]. Thus, the importance of HDL transporters in ocular VK biology are implicated.

Yamanashi et al. (2017) [79] have recently linked the intestinal absorption and export of dietary PK by enterocytes to membrane proteins previously known to be essential in cholesterol metabolism, including Niemann-Pick C1 like 1 (NPC1L1), ATP-binding cassette transporter A1 (ABCDA1), scavenger receptor class B type 1 (SR-B1), and cluster of differentiation 36 (CD36). These proteins are expressed in the retina [80], suggesting the retina maintains internal lipid processing machinery that involves HDL-like particles of which PK and MK4 are likely components [81]. Genome-wide association studies have linked these HDL transporters and their receptors to carotenoid levels in the human retina [78], and the transporter ABCDA1 has also been linked to primary open-angle glaucoma risk [82]; these findings indicate that these transporters and their participation in VK biology need to be considered in further research if a more complete understanding of the role of VK in the EVS is to be realized.

Thus, fundamental gaps exist with respect to the forms, tissue distribution, concentrations, and trafficking of VK within the human eye and, to a lesser extent, the human brain. These studies are required and urgently needed prior to any well-designed clinical trials of therapeutic interventions, as well as subsequent solid recommendations with respect to optimized VK dietary intake or supplementation throughout the life cycle.

### 2.2. Vitamin K-Dependent Proteins in the Ocular/Visual System

Having established that VK is found in the mammalian eye, as well as in human, mouse, and rat brains, and that MK4 levels in human decedent brains are correlated with recognized dementia-related neuropathologies and cognitive function prior to death, it is appropriate to begin to address the nature of VK action in the EVS.

Broadly, VK function can be categorized as canonical, primarily referring to the gamma carboxylation of hepatic coagulation-related VKDPs and secondarily referring to the gamma carboxylation of non-hepatic VKDPs, and non-canonical, referring largely to VK quinones’ ability to share electrons in biologic systems.

It has been nearly fifty years since the mode of action of VK, i.e., the γ-carboxylation of glutamic acid, was unambiguously demonstrated in relation to its activation of VKDPs by Nelsestuen et al. in 1974 [83], which was then noted by Suttie in his excellent review of vitamin K-dependent protein synthesis in 1993 [30]. There are currently believed to be 17 VKDPs [33]. Interestingly, γ-carboxyglutamyl carboxylase (GGCX), which is abundantly found in brain tissue [84], as well as in the retina [85], has also been shown to be a VKDP [86], though it is essentially never mentioned in lists of VKDPs. Importantly, very few works regarding VKDPs in general mention these proteins in the brain, and even fewer with respect to the EVS.

In 1998, Ferland [32] reviewed the 13 then-known VKDPs and classified 7 as related to blood coagulation, 3 as related to bone metabolism, and 4 as “other” with largely unknown functions, apart from Gas6, which is a regulator of cellular growth. No mention of VKDPs in the eye or brain were noted.

Having conducted ground-breaking work on brain sphingolipids, as well as the relationship between VK intake and cognitive function in rats and humans, Ferland subsequently published a comprehensive review of the action of VK in the nervous system in 2012 [84], in which the roles of Gas6 in cell survival, growth, and myelinations and of Protein S in signal-mediated neuroprotection and local antithrombotic activity in the brain were discussed. Again, there was no mention of VK or VKDP action in the EVS.

In an expansive effort, McCann and Ames (2009) [18] systematically reviewed VK and VKDP within the framework of the “Triage Theory” of micronutrient distribution and allocation, which posits that “when the availability of a micronutrient is inadequate, nature insures that micronutrient-dependent functions required for short-term survival are protected at the expense of functions whose lack has only long-term consequences, such as the diseases associated with aging” [18]. This work is the most detailed coverage of the major known VKDPs, and though limited, it is the only review to mention extrahepatic VKDPs and their action in the eye, referencing prothrombin (Factor II) in the cornea, and the role of Tgfbi in corneal dystrophy (which recent research calls into question with respect to its activation by VK [87]).

Before addressing the VKDPs specifically found in the EVS, the importance of VK and the seven hepatic VKDP clotting factors with respect to hemorrhagic visual complication should briefly be mentioned. A properly functioning clotting system is essential for the preservation of both life and visual function. VK deficiency in the newborn period can result in blindness in survivors [88]. Children treated with vitamin K antagonists (VKA), such as warfarin, and not properly monitored have been blinded due to orbital hemorrhages [89]. Numerous cases in the literature report major ocular complications associated with the use of VKA in adults as well [90,91]. Thus, the disruption of hepatic VK metabolism from any number of causes has long been known to result in potentially blinding complications. We can now turn our attention to the lesser-known extrahepatic VKDPs and their localization and potential importance to the EVS.

To date, four VKDPs, along with GGCX, have been shown to have expression and activity in ocular tissues. Using technology that they largely pioneered and developed to identify and isolate differentially expressed genes from very small human tissue samples [92], in 2000, Borrás et al. [93] reported for the first time that the gene coding for the matrix Gla protein (MGP) was among the most highly expressed in human trabecular meshwork tissue. Later, in 2000, Canfield et al., after having studied the osteogenic potential of bovine retinal vascular pericytes (BRVP) for more than a decade, reported that MGP was differentially expressed by BRVP during the deposition of a calcified matrix [94]. Borrás et al. went on to carefully prove over the next two decades that MGP was essential to maintain physiologic intraocular pressure [95], as well as to show the presence of MGP gene expression in the trabecular meshwork (TM), in the sclera above the TM, in the ciliary muscle, adjacent to the optic nerve in the peripapillary sclera [96], and in the vascular smooth muscle cells of the retinal vasculature [97]. Most recently, in 2021, Sarosiak et al. [98] demonstrated the presence of MGP and active VK metabolic pathways in the corneas of normal controls and patients with Schnyder corneal dystrophy.

In 2003, Collett et al. reported that bovine retinal capillary pericytes (BRCP) produced Gas6 [99], and Valverde et al. (2004) found Gas6 in bovine vitreous humor [100]; meanwhile, in a series of papers between 2001 and 2005, Hall et al. demonstrated that both Gas6 and Protein S were essential for the phagocytosis of retinal outer segments (ROS) by the retinal pigment epithelium (RPE) [101,102,103].

Finally, prothrombin (Factor II) has been shown to be produced locally at all levels in human corneas and to participate in corneal wound healing [104]. Table 3 below summarizes the currently known ocular-associated VKDPs and their location, number of Gla residues, and proposed functions.

## 3. Vitamin K and Specific Ocular Conditions

### 3.1. Glaucoma

Glaucoma is the leading cause of irreversible blindness, with a calculated 3.6 million people blinded due to glaucoma in 2020, and an estimated age-adjusted prevalence of 2.04 (per 1000) worldwide [105]. Glaucoma is a multi-factorial ocular condition characterized by optic neuropathy and generally clinically associated with increased intraocular pressure [106]. The ongoing reduction in intraocular pressure remains the only proven treatment for the condition, and currently, there is no cure [107,108]. Intraocular pressure is determined by the resistance to the flow of aqueous humor generated by the trabecular meshwork (TM), located in the anterior chamber of the eye at the juncture of the iris and cornea [109], with most of the resistance believed to be a function of the modulation and turnover of the extracellular matrix (ECM) in the region [110,111].

The VKDP MGP has been conclusively demonstrated to play a central role in glaucoma. Borrás et al. (2020) [95] showed that the mouse *Mgp* gene is both responsible and sufficient to maintain normal pressure in the eye, though the exact mechanisms remain to be determined. This discovery was the result of meticulous work beginning more than two decades ago [92,93], demonstrating that the gene encoding MGP is among the 10 most expressed genes in the human TM and that MGP activity is correlated with the calcification process [112]. They also demonstrated that MGP is heavily expressed in the TM, regionally in the sclera overlying the trabecular meshwork, and surrounding the optic nerve [96].

A detailed recounting of the twenty-year effort and cardinal milestones in this research achievement from the Borrás lab is beyond the scope of this review but is a remarkable accomplishment and worthy of recognition—one that may well usher in a new era of glaucoma gene therapy, a welcome option for possible glaucoma treatment in the future.

No studies on humans have yet been conducted to directly investigate VK with respect to glaucoma. However, Wei et al. (2018) [113] demonstrated that circulating immature nonphosphorylated and uncarboxylated MGP (dp-ucMGP), a marker of reduced VK nutritional status, is a long-term predictor of reduced retinal arteriolar diameter. The authors also referenced a number of studies linking retinal microvascular traits and generalized arteriolar narrowing with glaucoma [113]. A single in vivo study exploring the effect of dietary PK in a surgically induced ocular hypertension rat model was found [114]. Deng et al. established that rats fed a high PK diet for a total of seven weeks were protected from retinal ganglion cell (RGC) loss and TM injury. MGP levels were supported, and there was a transient reduction in intraocular pressure (IOP) at two weeks. Serum PK levels were 0.4 ± 0.52 ng/mL vs. 7.62 ± 5.98 ng/mL; *p* = 0.054 in controls vs. the experimental group, which was a 19× elevation, yet no significant adverse effects were found on coagulation parameters or body weight.

In 2018, a systematic review and meta-analysis of the effects of vitamins on glaucoma, encompassing 629 publications, found only dietary vitamin A to have a significant association with primary open-angle glaucoma (POAG), with a pooled OR [95% CI] of 0.45 [0.30–0.68] (I^2^ = 0%) [115]. The authors noted that no studies specifically related to VK were found, but importantly, four studies related to the intake of green leafy vegetables(recently shown in a prospective randomized cross-over clinical trial to cause a statistically significant improvement in VKDP parameters [67]) demonstrated significant negative associations with POAG. Increased dietary intake of green leafy vegetables was associated with a 20–69% lower risk of glaucoma [116,117,118,119]. Though the quality in the studies varied, the results suggest that even a relatively modest rise in one’s intake of green leafy vegetables, which are the main source of VK in the diet, may have a meaningful impact on the risk of glaucoma. Table 4 below provides a summary of the studies.

Thus, recent work confirms the role of VK with respect to glaucoma and supports the need for further research. A mouse model confirms the essential function of the VKDP MGP in glaucoma which may serve as a powerful platform for future VK ocular research. In vivo studies in rats have shown that PK may be effective in preserving RGC cells and addressing TM injury. Several observational studies have found that increased VK intake may reduce the risk of glaucoma, and that systemic VK status is associated with retinal arteriolar narrowing, a risk factor associated with glaucoma.

### 3.2. Cataracts

Cataracts are the leading cause of reversable blindness worldwide [120], with a global age-standardized pooled prevalence estimate (ASPPE) of 17.20% (95% CI 13.39–21.01) [121], indicating that for 1000 people randomly sampled anywhere, 133–210 are estimated to have cataracts. Age and regional differences were found to correlate with cataract type directly, and age was the source of variance for cataract prevalence [121]. This is not surprising given the fact that fetal lens cells persist for the lifetime of an individual [122] and are felt to serve as an excellent model of the aging process.

Despite advancements, unoperated cataracts remain ophthalmology’s major unsolved problem [123], and efforts to train and support additional surgeons cannot keep pace with aging demographics [120]. Measures to prevent or treat cataracts have met limited success, but antioxidants and free radical scavengers have shown potential in the lab, and thus further research and clinical trials are needed [124].

In a secondary analysis, Camacho-Barcia et al., 2017 [15], analyzed prospectively collected PK intake using baseline and yearly food frequency questionnaires in a group of 5860 individuals in the Prevención con Dieta Mediterránea Study (PREDIMED) [125], with a mean (SD) age of 66.3 (6.1) years, followed for a median of 5.6 years. After adjusting for multiple potential confounders, individuals in the highest tertile of mean energy-adjusted dietary PK intake had a lower risk of cataract surgery than those in the lowest tertile (HR, 0.71; 95% CI, 0.58–0.88; *p* = 0.002). A total of 768 individuals underwent cataract surgery. Notably, the European Food Safety Authority has set an adequate intake (AI) of 1 μg phylloquinone/kg body weight per day for all age and sex groups [48], and though the PREDIMED study did not report body weights, the average weights for adults in Spain are 82 kg and 66 kg for males and females [126]; the United States and Canada AI-s are 120 and 90 μg/day [127]. Thus, on average, the estimated daily intake of PK in this cohort was 2.75–5.5 times the recommended AIs on the low and high ends respectively, yet a near doubling of the PK intake from 249.4 to 496.7 µg/d was associated with a 29% reduction in the hazard ratio (HR) for cataract surgeries comparing the lowest to highest tertiles. A review of the supplemental data reveals that daily average green leafy vegetable intake was 48.2 g/day and 130.2 g/day in the two groups, representing a difference of approximately 1/2 cup of cooked kale per day [127].

Though experts have pointed out a number of limitations in the study methodology with respect to the estimation of dietary PK intake [40], the study population size, controlled for multiple confounding dietary and other variables, the firm end point of a large number of surgeries, along with the 29% relative risk reduction in such a clinically relevant indicator, that occurred in close to 1 in 8 participants, may well serve as a strong rational for the undertaking of well-designed interventional studies with respect to VK and cataract prevention. Notably, and possibly strengthening this assertion, is a companion study by the authors [128] reporting on these same individuals, showing that the incidence of cataract surgery was similar in the interventional MedDiet groups supplemented with olive oil or nuts, compared with the low-fat diet group, suggesting that a doubling of PK intake itself may be protective, while other more general dietary factors are not. Certainly, well-designed interventional studies will be needed to define possible causality.

In a series of animal studies beginning in 2014, Varsha and Thiagarajan et al. [129,130,131], in a streptozotocin (STZ)-induced diabetic model in Wistar rats, demonstrated that PK was able to prevent the formation of lens opacity by modulating lens Ca^++^ homeostasis and hyperglycemic effects through direct action on the pancreas. In a second experiment they found that PK was able to reduce rat lens sorbitol via direct competitive inhibition of DL-glyceraldehyde binding to lens aldose reductase 2 (ALR2). Lens calcium in PK-treated rats was lower than that in STZ treated rats and even slightly lower than the control. Likewise, Ca^++^-ATPase activity was higher in PK-treated rats compared to the STZ group and controls. This suggested to the authors that PK has the potential to inhibit diabetic cataracts by supporting pancreatic function, inhibiting lens oxidative stress, and directly modulating lens Ca^++^ levels. Figure 4 shows the PK inhibition of STZ-induced diabetic-cataracts in Group III from Sai Varsha et al., 2014 [129].

The human lens is the only transparent, avascular, and nerveless organ in the body [132], and Ca^++^ ion concentrations, which are reduced 10,000-fold across the lens plasma membrane compared to the surrounding aqueous humor, have long been known to be a key factor in lens physiology and pathophysiology, largely serving as secondary messengers to regulate numerous essential cellular processes [133]. Receptor tyrosine kinases (RTK) and G-protein-coupled receptors are the cell plasma membrane receptors that detect extracellular signals and modulate free Ca^++^, thus controlling cell processes such as gene expression, cell proliferation, apoptosis, and cell death [133]. Gap junctions between lens cells serve as channels for the flow of metabolites, small molecules, and ions in the avascular tissue [134]. Mutations in the connexin proteins making up these junctions disrupt the calcium ion flux, and in extreme cases, lead to intracellular calcium precipitation and cataract formation [134].

Tyros3, Axl, and Mertk are RTKs that make up the TAM family of tyrosine kinase receptors that have Gas6 and PS, two VKDPs, as their ligands, with important function in the nervous system [135] as well as in the retina, as demonstrated by Hall et al. [103]. Valverde et al. (2004) [100] have confirmed Axl expression in rat and bovine lens epithelial cells, as well as in human lens epithelial cells. They also verified the presence of Gas6 in bovine aqueous humor. Using a human lens epithelial cell culture model, they show that Gas6, through Axl activation, supports lens epithelial cell growth, survival, and homeostasis. Interestingly, Carnes et al. (2018) report elevated *Gas6* mRNA expression in human ciliary bodies compared to human trabecular meshwork and corneal tissue, suggesting that this important VKDP ligand may be exported into the aqueous humor, in part to support lens physiology [136]. PS has been found free in aqueous humor in the 7–8 ng/mL range and varies in disease states such as diabetes [137], potentially indicating an additional role for VK in homeostatic mechanisms related to diabetic cataracts.

### 3.3. Schnyder Corneal Dystrophy

The eye, and particularly the cornea, have played pivotal roles in the elucidation of both the existence, and function, of fat-soluble vitamins, beginning with vitamin A and its role in xerophthalmia and keratomalacia [138,139] and extending to the essential role of vitamin A in night blindness and visual function [140]. Some of the earliest references to night blindness being treated with vitamin A-rich liver appear in ancient Egypt and Greece, with the bulk of the scientific work with regard to vitamin A being completed in the late 19th and early 20th centuries [140].

In the 21st century, the cornea provides a window into the physiology of VK with respect to the visual system and, more broadly, into complex intracellular molecular interactions in general [141]. This work is largely the result of the completion of the Human Genome Project [142] and genomic advances driving basic and translational research, resulting in an enhanced ability to identify and study genetic variations underlying diseases [143,144,145].

*UBIAD1* is the gene responsible for Schnyder corneal dystrophy (SCD), a rare autosomal dominant disorder that manifests as a progressive bilateral corneal opacification due to the widespread deposition of intercellular and extracellular cholesterol and phospholipids in the epithelium, Bowman’s layer, and stroma of the cornea [146,147,148]. It is variably associated with systemic abnormalities in serum lipids, lipoproteins, cholesterol levels, and, most recently, higher serum immature dp-ucMGP, a mark of VK deficiency [98].

The modern era of VK research with respect to the eye can be said to have begun in July 1987 with the referral of a 56-year-old “Swede-Finn” woman to Jane S. Weiss, M.D., at the University of Massachusetts Medical Center, for consultation regarding her cloudy corneas [149]. She was diagnosed with what was known at the time as Schnyder’s crystalline corneal dystrophy (SCCD), and within 18 months, two other individuals, possibly related to the same family, were also identified. A “unique opportunity to study a large number of patients with Schnyder’s crystalline dystrophy” was identified, and in what was otherwise considered a rare disease, a systematic effort was launched in January 1989 to undertake its study [150]. Between 1990 and 1992, a series of papers were published detailing the clinical and laboratory findings, epidemiology, genetics, lipid profile, histology, histochemistry, and ultrastructure, of one hundred and seventy-three living members of four families, all of whom originated from a 100 k^2^ area in southwest Finland near the Bay of Bothnia [147,149,151,152].

In 2007, this work culminated in the independent identification of the UbiA prenyltranferase domain-containing protein 1 (*UBIAD1*) gene on chromosome 1p36 as the cause of the disease by three groups employing various techniques [153,154,155]. However, the function of UBIAD1 and its relationship to corneal cholesterol deposition remained unknown.

Using the human genome database to screen for relevant prenylation enzymes, Nakagawa et al. (2010) demonstrated that UBIAD1 was a novel and ubiquitously expressed human enzyme that showed activity in most tissues and was responsible for the biosynthesis of MK4 within cells [156]. UBIAD1 was shown to have enzymatic activity to cleave the prenyl side chains from PK to generate MD and to possess prenylation activity to convert MD to MK4 in proportion to the concentration of the prenyl side group, geranylgeranyl pyrophosphate (GGPP) [156]. The authors concluded that both MK4 and UBIAD1 were likely physiologically essential cellular factors, and that the biosynthesis of MK4 by UBIAD1 may be related to the pathophysiology of SCD, though the mechanism was unknown [156]. In 2013, Nickerson et al. demonstrated that MK4 synthesis was significantly reduced by 22 to 39% in individuals with three different *UBIAD1* mutations compared to wt UBIAD1 and that UBIAD1 interacted with the enzymes that catalyze cholesterol synthesis and storage, 3-hydroxy-3methylglutaryl-CoA reductase (HMGCR), and sterol O-acyltransferase (SOAT1) [157]. This suggested to the authors that impaired MK4 synthesis is the biomolecular defect in SDC patients and that UBIAD1 links VK and cholesterol metabolism, with endogenous MK4 playing a role in sustaining corneal health and visual acuity [157].

In 2015, DeBose-Boyd et al. began to explore how UBIAD1 may contribute to excess cholesterol accumulation in patients with SCD and the novel link the between synthesis of MK4 and cholesterol. This resulted in the discovery that SCD UBIAD1 mutants are sequestered in the endoplasmic reticulum (ER) and bind to HMGCR, which is the rate-limiting enzyme for cholesterol synthesis, preventing its normal endoplasmic reticulum-associated degradation (ERAD) and leading to increased and dysregulated production of cholesterol and nonsterol isoprenoids [141,158,159,160,161,162,163]. This work, stimulated in part by SCD, has led to a much deeper understanding of the regulation of the mevalonate pathway, with implications not just for cholesterol metabolism, but also for a number of other essential molecules including steroid hormones, vitamins D and K, bile acids, Heme A, ubiquinone, and other nonsterol isoprenoids, as well as in cardiovascular diseases and cancer [141].

Dong et al. (2018) [164] reported the creation of a CRISPR/Cas 9-based *Ubiand1* mutant mouse with an N100S missense mutation which corresponds to the most common N102S variant reported in humans. The authors noted that mitochondrial dysfunction is a prominent component of the disease and suggested that this could result in dysregulated cholesterol synthesis and the accumulation of cholesterol in SCD corneas, but they found no differences in corneal cholesterol levels and there was an absence of cholesterol crystals among the control and mutant genotypes mice. Thus, the authors concluded that the implications of VK deficiency may be of particular importance in the cornea, stemming from the unique conditions present there, as well as the possible differences between humans and mice with respect to cholesterol metabolism [164].

In 2018, Sarosiak et al. [165,166] began reporting important clinical and molecular findings on four previously unpublished Polish families with SCD that were followed prospectively over at least seven years, of which one member presented a novel UBIAD1 variant, UBIAD1 p.Thr120Arg. In a 2021 follow-up study designed to explore the corneal and vascular VK status in SCD patients by focusing on matrix MGP, Sarosiak et al. [98] reported for the first time, “quite unexpected and remarkable findings” related to MGP expression in human corneas, along with differential perturbations by SCD UBIAD1 variants of associated VK metabolic pathways.

They showed that MGP is abundantly present in its fully activated form in SCD and normal control corneas, noting that this was quite unexpected given the known reduction in MK4 synthesis associated with UBIAD1 pathologic variants. They document high levels of post-translationally modified phosphorylated and carboxylated, active mature MGP (pcMGP), throughout the corneal epithelium. This pcMGP was localized in both the cytoplasm and nuclear compartments, and it was stratified and most highly concentrated in the epithelial basal layers of both the SCD and normal corneas. They further hypothesize, as others have done [55], that external and internal corneal specific factors such as mechanical forces, inflammation, and pH, may influence calcium levels differentially, and thus explain the stratification of the MGP. They note however, that there have been no reported calcium deposits in SCD corneas, and that they found no calcium staining in these cases when specifically searched for. They conceded that given the multifunctional nature of other VKDPs such as PS, and Gas6 [167,168], MGP may have some yet unknown specific function associated with corneal intracellular calcium homeostasis, as suggested by Proudfoot et al. [169].

MGP was also located in the corneal stromal keratocytes with no detection in the extracellular matrix. Using light microscopy, no immature ucMGP and dpMGP were detected within the corneal epithelium in either SCD or normal controls. The BGLAP gene encoding the VKDP osteocalcin was not found in SCD patients or controls, and no calcium deposits were detected.

UBIAD1 variants differentially increased MGP mRNA 12.31-fold in p.Asn112Asp patients and 3.25-fold in p.Asp102Ser. GGCX mRNA was increased 2.38- and 1.48-fold respectively, while Vitamin K Epoxide Reductase Complex Subunit 1 (VKORC1) mRNA was increased 1.90-fold in p.Asn112Asp UBIAD1, however in the p.Asp102Ser UBIAD1 patient, no difference in VKORC1 levels was seen compared to controls. Thus, UBIAD1 variants were shown to influence VK related gene expression and do so uniquely. In this case, MGP mRNA differed by nearly a factor of 4x. The authors speculate these VK metabolism related genes are up regulated to compensate for the reduced GGCX carboxylation of MGP due to lowered MK4 cofactor generation, though they acknowledge that the mechanism remains unclear, and that how this might relate to SCD phenotypic expression is unknown.

Lending support to these findings and possibly supplying corroboratory data, Chen et al. [170] reported a HEK293 cell-based model in which they selectively inserted and tested UBIAD1 mutant variants and independently measured MK4 biosynthesis and VKD protein carboxylation. Comparing p.Asn112Asp to p.Asp102Ser variants, they found an approximate 4.7× difference in VKD carboxylation which is in line with the nearly 4× MGP mRNA induction differences produced by these variants found by Sarosiak et al., thus showing some proportionality in the observed variant effects. Chen et al., commenting on their findings, noted that SCD patients do not in general exhibit typical phenotypic expressions associated with defects in MK4 or VKDP under carboxylation, and that their findings suggest that UBIAD1’s MK4 biosynthetic activity does not correlate directly with SCD patient corneal phenotypic expressions.

Sarosiak et al. also measured serum dp-ucMGP, a biomarker of functional vitamin K status in SCD patients, unaffected family members, and unrelated controls, and found for the first time, that the levels were significantly higher in SCD individuals, reflective of low systemic vitamin K activity.

The authors concluded that VK metabolic machinery is active in SCD and normal corneas, and that MGP appears to have a vital role in maintaining corneal health and possibly in protecting against corneal calcification. Sadly, despite these major advances, they acknowledged, as others have, that the “molecular mechanism by which UBIAD1 pathogenic variants affect the cornea leading to lipid deposition in SCD patients has yet to be determined” [166].

### 3.4. Retinal Disease

Unlike glaucoma, cataracts, and SCD, specific research with regard to VK and retinal disease are more limited, though this is likely to change quite rapidly with the recent discoveries with regard to VK and ferroptosis [26,171] addressed below. Age-related macular degeneration (AMD) is the most common cause of irreversible blindness, accounting for 8.7% of global cases [172], with an estimated prevalence of 196 million in 2020, increasing to 288 million by 2040 [173]. While AMD represents the largest well-phenotyped cohort of people who have provided dietary intake data examining the relationship of diet and nutrition with visual function [174,175], the association with VK has not been addressed, and the lack of findings regarding VK and the retina is not surprising, given that a search of the 433,444 trials listed in the U.S. National Library of Medicine Clinical Trials database with the search terms “vitamin” and “retina” returned 83 trials [176] and zero with the use of to “vitamin K” and “retina” [177]. A few human and basic research studies are, however, available.

Wei et al. (2018) [113,178] reported in a large prospective longitudinal study that a doubling of serum-inactive VKDP dp-ucMGP, an inverse serum marker of systemic VK status [179], was associated with a 1.40 μm narrowing of the retinal arteriolar diameter and concluded that their findings underscore the pivotal role of activated MGP in ocular homeostasis and that VK supplementation may promote retinal health [113]. The observations were further thought to be clinically relevant because smaller retinal arterioles and lower arteriole-to-venule diameters predict cardiovascular mortality [180] and coronary heart disease [181] and may be reflective of similar changes in the cerebral microvasculature [182], thus linking MGP activity in the eye to the broader topic of VK in aging and age-associated diseases [183,184,185]. Borrás et al. likewise confirmed the expression of MGP in mouse retinal vasculature, including capillaries and pericytes [96,97].

Years earlier, beginning in 1990, Canfield [186] and others at Manchester University, began to report on the spontaneous transition of bovine retinal microvascular pericyte cells (BMPC) into multicellular nodules that produced hydroxyapatite needle-like crystals (the deposition of calcium phosphate salts in the form of hydroxyapatite is the hallmark of vascular calcification [187]) and were ultimately differentiated along osteogenic lines. In 1998 [188], they reported, for the first time, that microvascular retinal pericytes have osteogenic potential in vitro and in vivo and are able to differentiate into bone, cartilage, adipose, and fibrous connective tissue, concluding that pericytes may give rise to cells of multiple lineages and that microvascular pericytes may play an important role in the calcification of vascular tissue. In 2000 [94], they showed that the VKDP MGP had a key role in regulating the differentiation of these multipotent cells as well in the calcification process itself.

Three years later, in 2003 [99], the same authors were the first to show that the VKDP Gas6 and its ligand, Axl, were highly expressed in cultured BMPC, and they suggested that Gas6/Axl signaling was important in the regulation of retinal physiology and pathophysiology. Then, in 2007 [189], the group was again the first to show that Gas6/Axl signaling inhibited mineral deposition by VSMCs in cultured vascular smooth muscle cells (VSMC) from bovine aortas, meaning it is potentially an important regulator of vascular calcification.

The work of Canfield et al. demonstrates the important and broad involvement of vitamin K with retinal vasculature and provides a strong mechanistic rationale to explain the findings of Wei et al. [113,178] with respect to the association of dietary vitamin K intake, dp-ucMGP, and reduced retinal arteriolar diameters. Therefore, as not to be remiss, it is important to point out, as Sweeney et al. (2016) [190] and (Simó et al.) 2018 [191] have shown in their excellent reviews, that the microvascular retinal pericytes [191], as well as pericytes of the brain and central nervous system [190], do much more than serve to support the vascular system: they have a critical role at the center of the neurovascular unit, which is made up of vascular cells, glial cells, and neurons. Thus, these findings link VK through MGP and Gas6, not only to the retinal vasculature but also to the neuroretina.

The regulation of the phagocytic function of the retinal pigment epithelium (RPE) is critical to photoreceptor function, and thus to vision, with the disruption of RPE phagocytosis leading to diseases such as retinitis pigmentosa and AMD [118]. Two VKDPs, Gas6 and PS, are critical ligands in this Mer receptor-mediated process [135]. Beginning in 2001, Hall et al. [101,102,103] demonstrated for the first time that both Gas6 and PS, another coagulation-related molecule, had a specific function in the eye. It was shown that Gas6 and PS require VK-dependent gamma-carboxylation before linking to the outer segments (OS) of shedding photoreceptors (PR) in a Ca^++^-mediated manner, and subsequently to activate RPE Mer receptors stimulating OS phagocytosis. Seven percent of the photoreceptor mass is said to turn over daily, making the RPE the most active and among the most specialized phagocytes in the human body [192,193]. Both Gas6 and PS are widely expressed in the nervous system and are involved in the regulation of phagocytosis in several important processes [102], in addition to their function with respect to the RPE. After having demonstrated the common and redundant roles of Gas6 and PS for the first time in any biological process, Hall et al. emphasized the critical role of VK and VKDPs in retinal OS phagocytosis, and thus in global retinal function [103].

## 4. Non-Canonical Vitamin K Functions and Ferroptosis

In what may prove to be the most significant advancement with respect to VK and the EVS in the last decade, reduced VK, apart from its interaction with VKDPs, has been shown to be a potent suppressor of a programmed cell death process known as ferroptosis [26,194,195].

VK’s ability to act as an antioxidant has been reported in the past [196,197], though it has long been held [19,61] that the only firmly established physiologic function of VK is its action as a cofactor for γ-carboxyglutamyl carboxylase (GGCX) to influence the post-translational modification of VKDPs. Recent discoveries, however, with respect to VK and its relationship to the ferroptosis suppressor protein 1 (FSP1) [26,171] may warrant a re-evaluation of this view.

A total 330 billion human cells are estimated to turnover daily (4 million per second) [198], likely the result of numerously evolved programs of regulated cell death (RCD) to control cancer [199] and promote cooperation for the survival of the organism [200]. Corneal epithelial cells, for example, are estimated to turn over every seven days [201]; yet, uniquely, and by way of contrast, the adult lens nucleus is comprised of cells generated in utero at approximately the sixth week of gestation [202], with no turnover of the cells or their membrane lipids [203,204], and they are present for a lifetime. This may in part be the result of adaptations in human lens cell membranes to contain high levels of sphingolipids (for which VK has been shown to have a direct role in their synthesis [205]), which are more resistant to oxidation and might confer unique chemical and physical stability to the lens fiber cells [204,206].

Our understanding of the orchestration of cellular turnover, and how cells live and die together [207], has undergone a profound transformation over the past one hundred or so years, as the mechanisms of RCD (death programs controlled by dedicated molecular machinery which are thus potentially modifiable through environmental, nutritional, pharmacologic, or genetic inputs, among others [208,209]), as opposed to accidental cell death (ACD; the sudden and catastrophic death of cells as a result of exposure to severe physical, chemical, or mechanical insults resulting in the disassembly of the cells plasma membrane [208]), have been discovered and elucidated. Cell death, once taken for granted as an inevitable and natural consequence of cell life, was first recognized as an essential function in directing the metamorphosis of reptiles in the mid-19th century and was further appreciated in the early 20th century as a genetically determined process in insects and amphibians [207].

Ferroptosis was first described by Dixon and Stockwell et al. in 2012 [210] as a non-apoptotic RCD dependent on intracellular iron which was genetically, biochemically, and morphologically distinct from apoptosis, necrosis, and autophagy. In a 2022 comprehensive review, Stockwell [211] defined ferroptosis as a form of cell death characterized by the iron-dependent accumulation of lethal membrane-localized lipid peroxides which sits at the intersection of reactive oxygen species (ROS) chemistry, metabolism, and iron regulation, with highly relevant biologic and therapeutic significance [211].

In 2019, Doll et al. [212] and Bersuker et al. [213] independently discovered that reduced ubiquinol (CoQ10), a lipoquinone related to VK menaquinones [214], mediated the suppression of ferroptosis in conjunction with its NADH-dependent oxidoreductive enzyme, FSP1. The myristoylation of FSP1 was shown to recruit the enzyme to plasma membranes, where it serves to reduce CoQ10 [213], which then acts as a lipophilic-radical-trapping antioxidant (RTA) to stop the production of lipid peroxides, demonstrating the NADH/FSP1/CoQ10 relay as a powerful suppressor of ferroptosis [212].

In a major advancement with respect to research on both VK and ferroptosis, Mishima et al. (2022) [26], extending their earlier work [212], discovered that the reduced form of VK (VKH2), also acting as a potent RTA to inhibit phospholipid peroxidation, possessed powerful anti-ferroptotic activity and, importantly, that FSP1 was the warfarin-insensitive VK reductase sustaining this reaction [26]. They thus established that FSP1 acts to reduce both CoQ10 and VK to RTA’s and that FSP1 is the long sought after antidotal enzyme to overcoming warfarin poisoning [215,216,217]. Independently, Jin et al. (2022) [171], using genome-wide CRISPR-Cas9 knockout screening, likewise demonstrated FSP1 to be the elusive warfarin-resistant vitamin K reductase responsible for the reduction in VK, establishing VK not only as a cofactor for VK-dependent carboxylation but also as a scavenger of phospholipid radicals to suppress ferroptosis [171].

The implications of these findings with respect to VK and the EVS are significant. Zhang et al. (2022) [218] provided a comprehensive review of ferroptotic biology and its involvement in major ocular diseases and concluded that the effects of ferroptosis cannot be underestimated, deeming ferroptosis-related cell death to be a critical emerging field of study [218]. Yang et al. (2022) [219] reported evidence in human primary RPE cells and in mice that the FSP1-CoQ10/VK-NADH pathways inhibit retinal ferroptotic pathology and may have a potential therapeutic role, supported by biochemical, histologic, and electroretinogram data [219]. Likewise, Wei et al. (2021) [220] offered experimental evidence that human lens epithelial cells and mouse lens epithelium are highly susceptible to ferroptosis and that cataractous and aging human lenses may exhibit more hallmarks of ferroptosis than any other human organ [220].

Though researchers are rightfully expressing caution and the need for clinical trials and focused research [218], these and related findings led Hirschhorn and Stockwell (2022) [195] to conclude that VK status may serve as a biomarker for ferroptotic sensitivity and propose that VK supplementation may be able to reduce symptoms of neurodegenerative and other ferroptosis-related conditions [195], which, by extension, likely include ocular and visual processing pathologies.

## 5. Vitamin K, Higher Cortical Visual Processing Function, and Ocular Correlates

Dietary VK intake, variations in gut MK isoform biosynthesis, and MK-4 concentrations in the brain are increasingly associated with cognitive function and neurodegenerative histopathology [63,66,221,222,223,224,225]. Though the domain of memory is often associated with, and evaluated as a measure of, cognition [226], it is important to recognize that visual perception, processing, and hallucinations are higher cortical functions as well and can be used as diagnostic criteria in neurologic degeneration [227].

Though memory tends to dominate cognitive neurology, it should be noted that visual function can be selectively impaired, and prominent cortical visual dysfunction, with relative memory sparing, can occur in various neurodegenerative conditions [228,229]. Increasingly, measurements beyond simple visual acuity assessment, such as the perception of biologic motion in mild cognitive impairment [230,231], as well as color vision and formed hallucinations in dementia with Lewy bodies [232,233], are being shown to be possible biomarkers of early disease. There is also a growing awareness of the possible cross-talk between decreased visual acuity and cognitive decline [234].

Likewise, the retina, by virtue of its common embryogenesis, as well as its anatomic and physiologic similarities to the brain, provides a unique “window” to observe correlations and potential consequences or brain pathology from the earliest to advanced stages, with increasing sensitivity, noninvasiveness, precision, and speed that are largely the result of advances in optical coherence tomography (OCT) [182,235,236].

Thus, not only does an understanding of VK biology have the potential to support and preserve vision, for example, by addressing cataracts and retinal pathology, but it also has the potential to have similar effects on cognition, and thus higher cortical visual processing-likely providing synergistic and compounding benefits to patients on multiple fronts. Targeting basic and transitional VK ocular research, as well as incorporating ocular and visual data and biomarkers in neurodegenerative research more broadly, can have significant impact, and should be considered a high priority given recent advances in the field [10].

## 6. Discussion

The eye plays a central role in the health and wellbeing of humanity and has long contributed significantly to unlocking important scientific discoveries [143,145].Vision and its loss has a particular immanency in the human psyche [1], and visual processing occupies a large percentage of the neocortex in terms of neural connections [53] and energy expenditure [237,238], largely a result of the high cost of moving ions across neuronal membranes, and the fact that the neuronal systems are a constant energy sink regardless of whether they are active or at rest [239]. Vision loss is an increasing problem worldwide as populations age, with significant economic and social impact [8] despite numerous advances in medical and surgical interventions [6]; therefore, it is imperative to identify and address the nutritional factors associated with maintaining and optimizing visual function [10,11]. Recent advances in VK research seem poised to provide much needed actionable insights.

It is well-established that the FSVs A, D, and E play essential roles and, in some cases, can have toxic effects in the eye and visual system [240,241,242]. Vitamin A, in the form of 11-cis-retinal, is indeed the singular light-sensitive molecule setting in motion the visual cycle, resulting in the transduction of photons into nervous impulses [241], and its deficiency has been heralded by night blindness and xerophthalmia for thousands of years, the direct result of which led to its discovery as the first FSV [138]. Vitamins D [240] and E [242], likewise, have been shown to have pleiotropic benefits as well as rare toxicity for the EVS, with toxic levels of A, D, and E, unlike VK, being able to accumulate within the body [243].

In contrast, VK tissue concentrations in mouse eyes and human brains are in the pmol/g range [55,62,63,66], several orders of magnitude less than its fellow FSV [19,64], and the human total body pool of VK is estimated to be as low as 17–194 μg (0.28–2.17 μg/kg), with a mean (SD) of 87.6 μg (60) and a total body pool turnover of approximately 1.5 days. This is a unique feature in the realm of FSV biology, and an important point to grasp when approaching problems and developing solutions related to this nutrient. These small concentrations and rapid tissue loss led directly to the discovery of VK in chicks, who are uniquely sensitive to the withdrawal of VK from their diet, showing clinical evidence of deficiency within hours. Similar vulnerabilities are seen in human infants, resulting in the nearly universal worldwide parenteral administration of VK shortly after birth to prevent vitamin K deficiency bleeding (VKDB) in early infancy [244], which is sometimes associated with vision loss. Also unique to VK is the fact that no known toxicity to naturally occurring VK isoforms has been shown, and natural forms of VK “appear to be essentially innocuous” [245]. Also noteworthy with respect to its fellow FSVs is the presence of naturally occurring VK antagonists in the form of the fungal conversion of plant-derived coumarins into dicoumarol, a prototypic anticoagulant [246].

Ample evidence now exists of the presence and importance of VK in the eye and visual system, primarily in the form of the menaquinone MK4, and to a lesser extent PK in the eye and human brain. The mouse gene *Mgp* has been shown to be both responsible for and sufficient to control physiologic mouse intraocular pressure [95]. Human brain MK4 concentrations are now known to be inversely associated with neurodegenerative histologic pathology and antemortem cognitive function loss [66], and a clinical trial has demonstrated a beneficial association of increased daily dietary VK intake and a reduced risk of cataract surgery [15]. However, there is a profound lack of human research and clinical trials with respect to VK and the EVS, which represents a significant opportunity.

To date, there appears to be no published data regarding the concentrations and tissue distribution of VK in the human eye. Additionally, age, gender, genetic, nutritional, microbiome, environmental, and regional factors affecting human ocular VK biology, have not been reported in any detail.

Despite the prominent importance vision plays in the human experience, and growing evidence of VK involvement in the EVS, not a single trial with respect to VK and the eye has been registered in the U.S. National Institutes of Health ClinicalTrial.gov database, and only one trial has been reported regarding VK and cataracts in PubMed [15]. The AREDS and AREDS2 studies, the largest and most well characterized trials related to macular degeneration, analyzed 44 nutrients, identifying 9, which were significantly (*p* < 0.0005) associated with decreased risk, and 3 with increased risk, of various forms of AMD [175]. Associations with VK were not reported however, though it is hoped that a secondary analysis of the existing data may provide insights regarding VK and AMD.

Several factors may explain this absence of clinical trials. First, there has been a general lack of awareness and appreciation for the potential of broad-based pleiotropic VK action in the eye. The dramatic and immediate clinical consequences of VK deficiency, and the utility of using VKA to treat blood clotting diseases, have dominated its nearly 100-year history [18]. This led directly to the early establishment of VK requirements based solely on the minimal amounts of PK needed to provide the liver with adequate PK to support functioning clotting factors [246]. Though PK and MK forms of the vitamin were discovered roughly in tandem, the abundance and importance of PK from plants in the human diet and the pressing need to prevent bleeding disorders, may have initially eclipsed any imperative to explore the extrahepatic requirements for VK. Ames and coworkers have advanced the “triage theory” [18,247] to highlight the tendency to focus on the life-threatening aspects of micronutrient deficiencies, at the expense of exploring the long-term debilitating consequences of critical nutrient inadequacy on age-related diseases, as exemplified, for example, by cataracts, glaucoma, and AMD.

Secondly, a lack of critical biomarkers, broad-based nutritional databases with respect to both PK and MK, and a comprehensive accounting of VK content and flux in the eye and brain, have all greatly hampered research interest and efforts. Furthermore, the assessment of VK status is made more difficult by the fact that serum VK levels have large interindividual variability, especially compared to other FSV, showing significant age, gender, racial and ethnic differences not solely related to intake, that remain largely unaccounted for [248,249,250], not to mention the fact that as many as 25% of participants in studies may have PK serum levels < 0.1 nmol/L (the current lower limit of detection) [250].

Finally, funding issues also have significant impact on VK ocular and visual system research. Though the categories are not mutually exclusive, The National Institutes of Health 2023 funding estimates for the category of Eye Diseases and Disorders of Vision are USD 1120 million, which ranks 54th in a list of 309 research/disease areas, contrasting with Alzheimer’s and Alzheimer’s Disease related dementias at USD 3324 million ranking 24th [251]. Likewise, the 2022 funding levels for the National Eye Institute was one-third of that of the National Institute of Neurological Disorders and Stroke for example, at USD 833 million vs. USD 2503 million [252].

Despite the recognition that nutrition is the most powerful intervention to reduce disease across the life span, and that the correction of micronutrient deficiencies yields substantial returns on investment [10], vitamin research in general, and VK research specifically, has lagged. In 2017, Chambers et al. reported that US federal funding for vitamin research declined between 1992 and 2015 as a percentage of total federal research grant spending [13], from 0.6% to 0.2%, or to USD 95 million (2016 US dollars). Between the years 2000 and 2015, VK research had the fewest average projects funded at 8/yr., compared to 115/yr. for vitamin A, with VK garnering the lowest average yearly project value of USD 2.4 million, versus USD 34 million for vitamin D, both of which represent a 14:1 ratio in favor of other FSVs. Increasing these funding levels, especially considering recent findings, has the potential to deliver needed breakthroughs to advance EVS nutritional recommendations and health.

Great strides have been made in the last 25 years to increase our understanding of VK biology, particularly in the areas leveraging the Human Genome Project, CRISPR/Cas9 technology, radioisotope studies regarding PK and MK, and the discovery of UBIAD1, to name a few. Most recently the role of VK in ferroptosis, is a major advance which is likely to rapidly drive VK research in the EVS for years into the future, and possibly serve as a much needed biomarker of VK status [195]. Also important are the prospectively gathered data indicating that increased dietary VK is associated with a decreased risk of cataract surgery, and that higher MK4 levels in decedent brains are inversely associated with a broad range of neurohistopathologies, and better overall cognitive function, all of which strengthen the nutritional epidemiologic evidence necessary to justify VK interventional trials [43,253].

## 7. Clinical Considerations

Clinical considerations with respect to VK and the EVS for those seeking consultation with an eye care professional might be reasonably drawn from the human studies cited. One can consider that the most recent analysis of the National Health and Nutrition Examination Survey (NHANES) data indicating that VK intake is declining and more than half of adults over 70 years of age are not meeting minimal dietary recommendations, which are 90 μg/d for females and 120 μg/d for males, [47]. Secondly, there are no known toxicities associated with VK intake in healthy individuals [245,254]. Therefore, it would seem reasonable to encourage most patients to increase their intake of green leafy vegetables unless there are potential medical contraindications such as the use of a VKA (warfarin for example) or other dietary limitations. Patients with glaucoma, and those wishing to postpone cataract surgery if possible, should be encouraged, if they are able, to consume a cup of nutrient-dense green leafy vegetables such as kale, or broccoli, as well as foods rich in MK daily, as even modest increases in intake have been shown [116,117,118,119] to significantly reduce the risk of glaucoma progression in observational studies. Similarly, the difference between one-half and one cup of green leafy vegetables a day in the PREDIMED study was associated with a 29 percent risk reduction in cataract surgery over less than a six-year period [15]. Patients with neurodegenerative diseases such as mild cognitive impairment, Lewy body dementia, AD, or Parkinson’s disease should be asked about subtle visual hallucinations, or assessed for disruption of color vision, and in the absence of contraindications, they should be encouraged with the same dietary recommendations as those with glaucoma and cataract concerns. Longitudinal ocular imaging such as OCT and fundus photos, as well as markers of higher cortical visual processing, should be considered, in an effort to promote the detection of neurodegeneration in the earliest possible stages, and to inform clinical decision making and therapeutic interventions, as well as to promote interdisciplinary and coordinated patient care, and research [255,256].

Health care professionals should recognize the pervasive nature of nutritional and micronutrient deficiency and their impact on multiple individual, global, and economic indices, particularly for certain vulnerable groups such as the economically disadvantaged, pregnant women, young children, and the elderly [18,257]. Widespread deficiencies are particularly acute for VK as well as Vitamins A, D, and E [257,258], with half of adults over 70 years old not meeting adequate VK intake recommendations according to US NHANES surveys, as noted above [254].

## 8. Conclusions

VK has been shown to be unique among its fellow FSVs with respect to its relatively small tissue concentrations, limited and precarious total body stores, and lack of any known significant toxicity. Evidence has been presented from basic and clinical studies that VK is found in the eye and brain, that the VKDP-MGP is responsible for controlling intraocular pressure, and that, for the first time, prospectively obtained VK levels in human brains have been associated with histologic neuropathology and a slower decline in cognitive function, while the increased dietary intake of VK has been prospectively demonstrated to be associated with a significant reduction in the risk of cataract surgery.

Taken together, the relevance of VK biology with respect to the eye and the visual system can thus be confidently asserted. The virtual lack of human VK tissue concentration and distribution data, along with no firm understanding of the trafficking of VK to and within the eye, reveals a significant gap in our knowledge, as well as a tremendous opportunity—inviting scientists, clinicians, funding partners, and advocacy groups, along with patients, to rise to the occasion and shape a healthier, clearer, and brighter future together.

## Figures and Tables

**Figure 1 nutrients-15-01948-f001:**
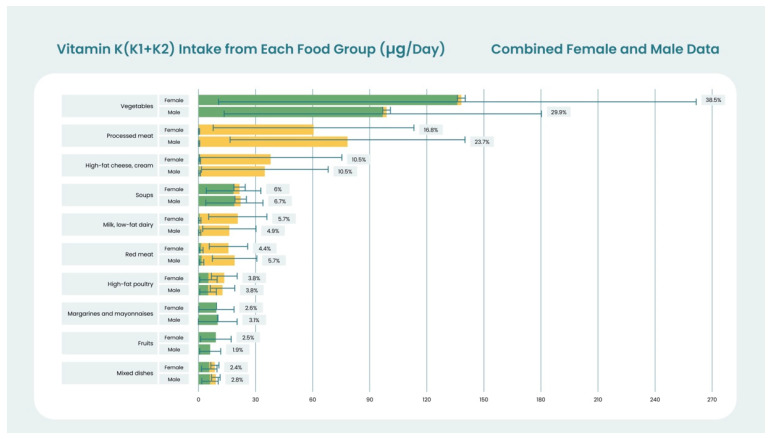
Combined female and male VK mean (μg/day) and SD daily intake in 1985 Polish adults aged 35-70 years, enrolled in the Polish arm of the global, epidemiological, cohort-based Prospective Urban Rural Epidemiology study (PURE) from 2007 to 2009. Values represent the top 10 of 25 food groups contributing 93% of the total VK dietary intake in both female and male subjects. Scale is 0–270 (μg/day), error bars represent SD for each food group by sex. Percent (%) values represent the percentage of total daily intake of VK by food group and sex. Green color represents PK, yellow represents Mk-n. Data presented to visualize source and relative PK and Mk-n content in contemporary diet contrasted by sex. Adapted from [52]. (CC BY license https://creativecommons.org/licenses/by/4.0/).

**Figure 2 nutrients-15-01948-f002:**
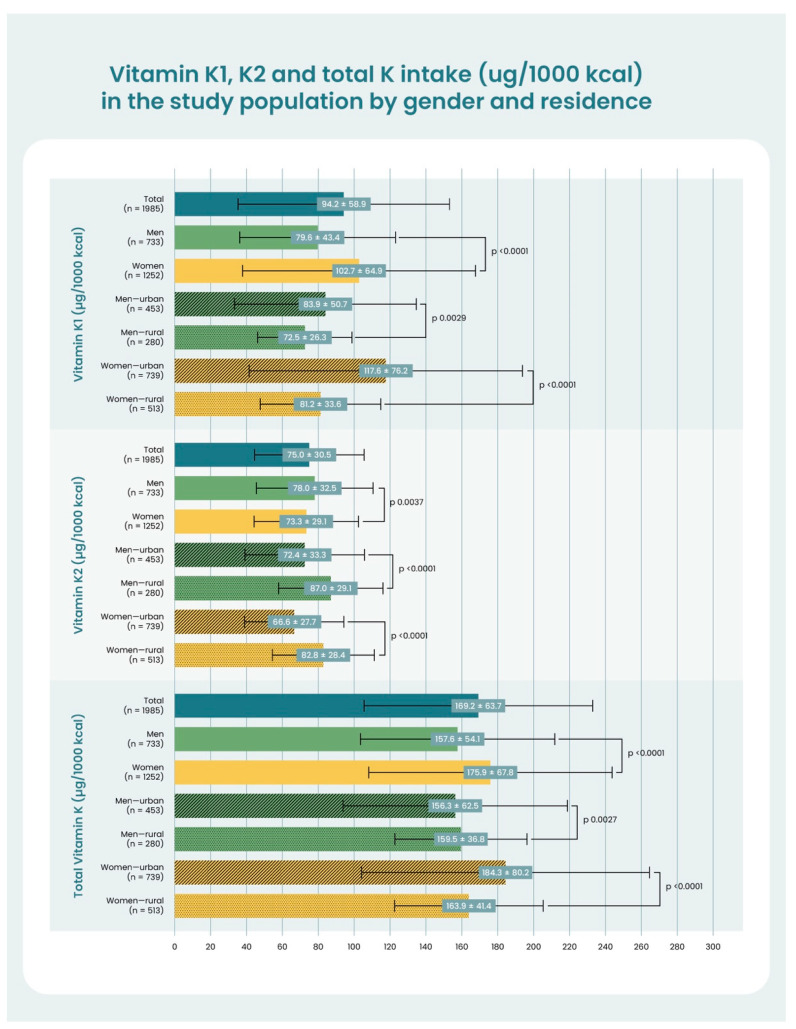
Vitamin K1 (PK), K2 (MK-n), and total VK intake (μg/1000 kcal) in the study population by gender and residence. Values represents the mean ± SD intake (μg/1000 kcal) in the study population and by sex and place of residence. *p* values represent results of Mann–Whitney U test comparison between groups. Data presented to aid in visualization of differences between female and male dietary patterns, as well as differences between urban and rural settings. Adapted from [52]. (CC BY license https://creativecommons.org/licenses/by/4.0/).

**Figure 3 nutrients-15-01948-f003:**
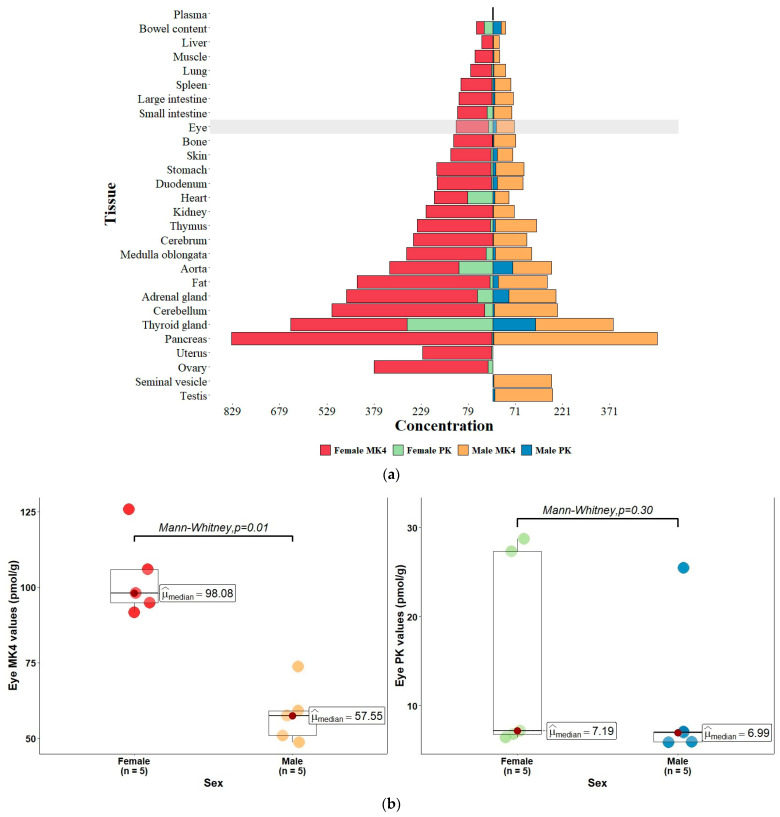
(**a**) Mean (*n* = 5) tissue MK4 and PK concentration (pmol/g tissue or pmol/mL plasma) in mice fed a normal diet sorted by female MK4 concentration. Red—female MK4; Green—female PK; Orange—male MK4; and Blue—male PK. Note tissue type, gender, female/male MK and PK ratios, as well as tissue type and gender MK/PK. (**b**) Box plots of MK4 and PK concentrations across sex in mice. Mann–Whitney U test used to obtain *p* values. MK4 *p* = 0.01, PK *p* = 0.3. Adapted from [55]. (CC BY license https://creativecommons.org/licenses/by/3.0/. Table 1 of original work reformatted to reflect relative eye MK4 and PK tissue concentrations. No endorsement is implied).

**Figure 4 nutrients-15-01948-f004:**
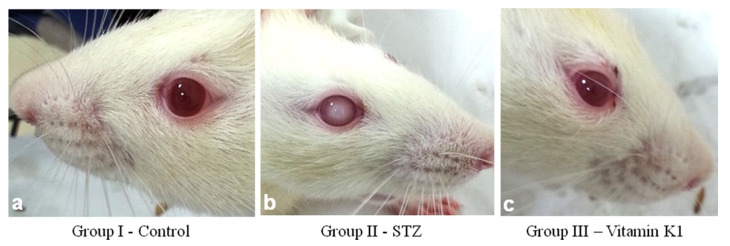
Effect of vitamin K1 on diabetic cataractogenesis. Control group I (**a**), cortical cataract observed in Group II STZ induced diabetes (**b**), treatment of diabetic rat with vitamin K1 prevented appearance of cataract in group III rats (**c**). Adapted from [129]. Copyright Elsevier-used with permission.

**Table 1 nutrients-15-01948-t001:** Common foods ranked by relative VK (μg) amount per typical serving for PK and MK4 from the USDA FoodData Central database.

Phylloquinone-Rich Food Description	Serving Weight (g)	PK (μg/100g)	Serving Measure	Serving PK (μg)
Spinach, raw	340	483	1 bunch	1640
Cabbage, cooked, boiled, without salt	1262	109	1 head	1370
Endive, raw	513	231	1 head	1180
Turnip greens, frozen, cooked, drained	220	519	10 oz package	1140
Broccoli, cooked	437	256	1 bunch cooked	1120
Collards, frozen, chopped, cooked	170	623	1 cup, chopped	1060
Parsley, fresh	60	1640	1 cup chopped	984
Kale, frozen, unprepared	284	334	1 package (10 oz)	947
**Menaquinone-4-Rich Food Description**	**Serving Weight (g)**	**MK (μg/100g)**	**Serving Measure**	**Serving MK4 (μg)**
Kielbasa, fully cooked, grilled	20.1	367	1 link	73.8
Pepperoni, beef and pork, sliced	41.7	85	1 serving (3 oz)	35.4
Butter, whipped, with salt	20.9	151	1 cup	31.6
Chicken, broilers or fryers, drumstick	35.7	85	1 serving (3oz)	30.3
Meatballs, frozen, Italian style	28.1	85	1 serving (3 oz)	23.9
Frankfurter, meat and poultry, cooked	35.6	48	1 frankfurter	17.1
Baby food, meat, turkey, junior	18.7	68	1 container	12.7
Baby food, meat, turkey, junior	18.7	68	1 container	12.7
Sausage, turkey, breakfast links, mild	36.6	27.9	1 link	10.2
Salami, cooked, beef and pork	28	12.3	1 slice round	3.4
Cheese, cheddar	9.3	105	1 cup	9.8

Source: https://fdc.nal.usda.gov/fdc-app.html#/?component=0 (accessed 18 March 2023).

**Table 2 nutrients-15-01948-t002:** Tissue distribution of VK in mice fed a normal diet. Values are presented as means ± S.E. (*n* = 5) in pmol/g tissue or pmol/mL plasma in female and male mice. *p* values represent results of Mann–Whitney U test comparison between female and male eye MK4 and PK groups.

Menaquinone-4 (MK-4)				Phylloquinone (PK)		
Tissue	Female (*n* = 5)	Male (*n* = 5)	Female:Male Ratio	Female (*n* = 5)	Male (*n* = 5)	Female:Male Ratio
			Eye tissue			
Eye	103.3 ± 6.1	58.0 ± 4.4	1.8 (*p* = 0.01)	15.3 ± 5.2	10.3 ± 3.8	1.5 (*p* = 0.3)
**Brain tissue**						
Cerebellum	487.7 ± 28.2	200.5 ± 17.5	2.4	26.2 ± 21.5	4.8 ± 1.8	5.4
Medulla oblongata	253.3 ± 5.5	116.2 ± 8.3	2.1	22.4 ± 15.5	7.6 ± 2.8	2.9
Cerebrum	252.5 ± 10.3	106.0 ± 8.2	2.3	1.1 ± 0.4	1.4 ± 0.7	0.8
**Other tissues in descending female MK-4 concentration**						
Pancreas	829.4 ± 56.7	520.1 ± 47.4	1.6	3.2 ± 0.4	3.0 ± 0.6	1.0
Fat	423.0 ± 60.1	155.2 ± 37.2	2.7	10.3 ± 2.1	17.4 ± 7.5	0.6
Adrenal gland	417.5 ± 139.6	148.6 ± 14.7	2.8	50.7 ± 10.6	50.7 ± 10.9	1
Thyroid gland	370.3 ± 64.0	247.3 ± 30.4	1.4	274.8 ± 140.3	134.9 ± 45.5	2.0
Ovary	363.4 ± 35.6		NA	16.2 ± 3.4		NA
Thymus	232.5 ± 12.0	131.2 ± 5.5	1.8	9.0 ± 3.9	7.5 ± 1.2	1.2
Aorta	220.0 ± 8.8	124.0 ± 13.7	1.7	109.2 ± 45.2	62.2 ± 32.6	1.7
Uterus	219.6 ± 31.8		NA	4.9 ± 1.2		NA
Kidney	212.7 ± 20.8	66.1 ± 4.9	3.2	1.7 ± 0.2	1.1 ± 0.1	1.5
Duodenum	172.8 ± 11.5	82.7 ± 6.0	2.1	5.0 ± 0.5	13.2 ± 8.8	0.4
Stomach	171.7 ± 14.4	90.4 ± 6.1	1.9	7.7 ± 1.1	7.8 ± 1.2	0.8
Skin	128.2 ± 26.2	50.2 ± 7.5	2.5	6.1 ± 1.3	12.7 ± 6.2	0.5
Bone	122.7 ± 10.2	69.8 ± 9.2	1.7	2.2 ± 0.5	3.2 ± 1.4	0.7
Heart	107.7 ± 10.0	46.4 ± 4.2	2.3	80.4 ± 76.3	4.9 ± 2.3	16
Large intestine	106.1 ± 13.1	60.1 ± 5.4	1.7	3.1 ± 0.7	5.5 ± 2.1	0.6
Spleen	100.6 ± 9.0	50.9 ± 5.0	2	3.9 ± 0.3	4.6 ± 0.9	0.8
Small intestine	93.6 ± 6.8	57.8 ± 7.8	1.6	19.7 ± 16.7	3.1 ± 0.6	6.3
Lung	67.1 ± 5.4	38.1 ± 4.2	1.8	5.5 ± 2.9	2.1 ± 0.4	2.5
Muscle	56.3 ± 7.5	19.1 ± 1.5	2.9	1.4 ± 0.2	2.2 ± 0.8	0.4
Liver	35.0 ± 2.2	18.2 ± 2.7	1.9	1.5 ± 0.2	1.2 ± 0.1	1.1
Bowel content	24.6 ± 4.7	13.6 ± 3.7	1.8	27.8 ± 2.1	24.9 ± 2.9	1.1
Plasma	1.2 ± 0.1	0.7 ± 0.1	1.7	0.6 ± 0.0	0.6 ± 0.0	1
Testis		185.3 ± 3.3	NA		5.5 ± 2.5	NA
Seminal vesicle		184.2 ± 9.1	NA		2.2 ± 0.6	NA

Adapted and modified from [55] (CC BY license https://creativecommons.org/licenses/by/3.0/. Table 1 of original work reformatted to reflect relative eye MK4 and PK tissue concentrations. No endorsement is implied).

**Table 3 nutrients-15-01948-t003:** Summary of ocular-related VKDP and their functions.

Author, Year Country [Ref]	Location	Protein [Number of Gla Residues]	Expression	Function
Sarosiak et al., 2021 Poland [98]	Human cornea	Matrix Gla Protein (MGP) [5 Gla]	Epithelium, stromal fibroblasts, and keratinocytes	Essentially unknown, possible maintenance of calcium homeostasis and anti-calcification role. Possible role in calcium signaling of epithelial proliferation and differentiation [52]
Borrás et al., 2000–2021 USA [93,95,96,97]	Human and mouse trabecular meshwork (TM), peripapillary sclera (SC), retinal vascular smooth muscle cells (VSMC)	MGP [5 Gla]	TM, SC, capillaries, and pericytes	Essential to maintain physiological IOP
Ayala et al., 2007 USA [104]	Human cornea	Prothrombin [10 Gla]	All layers of the cornea, intracellularly and in extracellular matrix	Production of thrombin, corneal wound healing, regulation of growth factors, and other signaling molecules via protease-activated receptors.
Hall et al., 2001–2005 USA [101,102,103]	Rat photoreceptor outer segments (POS), retinal pigment epithelium (RPE)	Growth Arrest Specific 6 (Gas6) [11 Gla], Protein S (PROS1) [11 Gla]	RPE, POS	Phagocytosis of POS by the overlying RPE via vitamin K-dependent gamma-carboxylation of Gas6 and PS in a Ca^++^-mediated linking of POS to RPE Mer receptors for phagocytosis
Valverde et al., 2004 USA [100]	Vitreous humor	Gas6 [11 Gla]	Bovine vitreous humor	Gas6/Axl signaling in lens epithelial cells support cell growth and survival.
Collett et al., 2003 UK [99]	Bovine retinal capillary pericytes	Gas6 [11 Gla]	Bovine retinal capillary pericytes	Interactions in regulating osteogenesis and diseases involving ectopic calcification.
Canfield et al., 2000 UK [94]	Bovine retinal capillary pericytes	MGP [5 Gla]	Bovine retinal capillary pericytes	Regulation of cell differentiation and calcification

**Table 4 nutrients-15-01948-t004:** Glaucoma and dietary intake studies.

Author, Year, Country [Ref]	Subjects	Design	Intervention	Findings
Coleman et al., 2008 USA [116]	1155 women with glaucoma in at least one eye	Observational Cross-sectional cohort	Block Food Frequency Questionnaire	Decreased glaucoma risk by 69% (odds ratio [OR], 0.31; 95% confidence interval [CI], 0.11 to 0.91) in women who consumed ≥one serving per month of collard greens and kale compared with those who consumed <than one serving per month.
Moïse et al., 2012 Africa [117]	500 African type II diabetics	Cross-sectional design	Mediterranean-style dietary score (MSDPS) using the Harvard semi quantitative FFQ adapted for Africa.	Regular intake of the Mediterranean Diet and the consumption of locally grown vegetables including Brassica Rapa, dry beans, Abelmoschus esculentus, and Musa acuminata significantly reduced the absolute risk of glaucoma.
Giaconi et al., 2012 USA [118]	584 African-American women participants in the Study of Osteoporotic Fractures	Observational Cross-sectional cohort	The Block Food Frequency Questionnaire	>1 serving/week compared to ≤1 serving/month of collard greens/kale decreased the odds of glaucoma by 57% (OR = 0.43; 95% CI: 0.21–0.85)
Kang et al., 2016 USA [119]	Prospective cohorts of the Nurses’ Health Study (63,893 women; 1984–2012) and the Health Professionals Follow-up Study (41,094 men; 1986–2012)	Prospective cross-sectional observational study	Primary exposure was dietary nitrate intake. Information on diet and potential confounders was updated with validated questionnaires	Very robust study demonstrating greater intake of dietary nitrate and green leafy vegetables was associated with a 20% to 30% lower POAG risk; and a 40–50% lower risk for POAG with early paracentral VF loss at diagnosis, in 1483 incident cases identified in 63,893 women and 41,094 men followed for more than 25 years.

## Data Availability

Not applicable.

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
