# Peer review of "Vitamin K and the Visual System—A Narrative Review"

_nutrients, 2023, doi:10.3390/nu15081948_

Round 1
Reviewer 1 Report
Vitamin K is an essential micronutrient widely used for the prevention and treatment of bleeding disorders, osteoporosis, and cardiovascular disease. It functions as a cofactor for gamma-glutamyl carboxylase, which catalyzes the post-translational modification of glutamic acid to gamma-carboxyglutamic acid (Gla) in a variety of vitamin K-dependent proteins, referred to as Gla proteins. Studies have shown that vitamin K can also function as an antioxidant to prevent oxidative injury and as an anticarcinogenic agent for a variety of cancers. Although many review articles cover the function of vitamin K in these fields, none of them cover the topic of vitamin K in the visual system. In this manuscript, Dr. Michael Mong systematically reviewed the importance of vitamin K in the visual system. The paper discusses: 1) vitamin K biology and the visual system; 2) vitamin K and specific ocular conditions; 3) non-canonical vitamin K functions and ferroptosis; 4) vitamin K, higher cortical visual processing function, and ocular correlates; and 5) clinical considerations. This review provides extensive and good coverage of the topic, which is beneficial for researchers in the field of vitamin K, nutrition, and ophthalmology. The author may consider the following comments for possible improvements.
General comments:
11. Table 1 presents the results of vitamin K tissue distribution in mice from reference 39 with additional information of female/male ratio. As the data in table 1 include different tissues and plasma samples, it would be more accurate to give the units as “pmol/g tissue or pmol/ml plasma” as in the original article. Additionally, the author used “0” to present data that was not in the original article, which may not be accurate since undetectable values or those equaling zero were indicated as ND in the original table.
22. The author should be consistent with significant digits. For example, the author states, “PK and MK4 values are consistently higher in females with a ratio of 1.49 for PK and 1.78 for MK4 in the eye” (page 5, lines 190-191). However, these two numbers appear to be presented as 1.5 and 1.7 in table 1, respectively. Additionally, the author appears to conclude that PK and MK4 values in the eye are consistently higher in females than males. However, authors of the original study concluded that, “Basically, tissue-distribution patterns of K-vitamins did not differ between male and female mice”.
33. It may be beneficial to add a statement, “However, the enzyme that cleaves the prenyl side chain of VK is unknown” after the statement “UBIAD1 was shown to have enzymatic activity to both cleave the prenyl side chains from PK to generate MD, and also to possess prenylation activity to convert MD to MK4 … …” (page 15, lines 603-606). Although the in vitro data in reference 143 demonstrated two enzymatic activities of UBIAD1, authors of this article concluded that, “These results indicate that UBIAD1 may cleave the side chain to release K3 and then prenylate it with geranylgeranyl pyrophosphate (GGPP) to form MK-4.” Additionally, a later review article (reference 24) by the leading author of reference 143 stated, “During intestinal absorption, a fraction of dietary phylloquinone is cleaved by an unknown enzyme(s) to release menadione,” suggesting that the cleavage enzyme is still unknown.
44. The statement, “… … a non-VKDP vitamin K cycle has been shown to be a potent suppressor of a programmed cell death process known as ferroptosis” (page 18, lines 765-767) is confusing. The author may want to say that the reduced vitamin K has been shown to be a potent suppressor of a programmed cell death process known as ferroptosis. In this regard, credit should also be given to Kolbrink et al. (Vitamin K1 inhibits ferroptosis and counteracts a detrimental effect of phenprocoumon in experimental acute kidney injury”, Cell Mol Life Sci. 2022;79(7):387. doi:10.1007/s00018-022-04416-w) who demonstrated that vitamin K suppresses ferroptosis.
55. In the sentence, “The VKDP mouse gene Mgp has been shown to be both responsible and sufficient to control physiologic mouse intraocular pressure” (page 21, lines 907-908), it is not clear whether a protein (VKDP, vitamin K-dependent protein) or the mouse mgp gene is the subject of the sentence.
Reviewer 2 Report
The authors provide an interesting review which brings into light the importance of vitamin K, given its essential role in physiological processes and in preventing eye diseases-related complications. I suggest making the following amendments to the manuscript:
1.The purpose of this paper is not clearly stated. Please clearly state whether you are looking for the difference between Vit K1 and Vit K2 or analyzing two forms of Vit K2 (MK-4, MK-7).
2.Please consider adding a table showing the food sources of vitamins K1 and K2.
3. Please cosider useing IMRAD structure (Introduction, Methods, Result and Disussion)
4. Table 1 and Fig 1 which are taken from another publication should not be presented here, a reference to the source and discussion will suffice.
